# J1: Incentivizing Thinking in LLM-as-a-Judge via Reinforcement Learning

**Chenxi Whitehouse**  **Tianlu Wang**  **Ping Yu**  **Xian Li**  **Jason Weston**
**Ilia Kulikov**  **Swarnadeep Saha**

FAIR at Meta
chenxwh@meta.com, swarnadeep@meta.com

## Abstract

The progress of AI is bottlenecked by the quality of evaluation, making powerful LLM-as-a-Judge models a core solution. The efficacy of these judges depends on their chain-of-thought reasoning, creating a critical need for methods that can effectively optimize this reasoning process. In this work, we introduce J1, a reinforcement learning framework for teaching LLM judges to think before making decisions. Our core contribution lies in converting all judgment tasks for non-verifiable and verifiable prompts into a unified format with verifiable rewards, enabling direct optimization of evaluation quality while mitigating positional bias. We then use RL to train thinking-judges at scales of 8B, 32B, and 70B and show that they obtain state-of-the-art performance across multiple benchmarks. In particular, J1-Qwen-32B, our multitasked pointwise and pairwise judge also outperforms o1-mini, o3, and a much larger 671B DeepSeek-R1 on some benchmarks, while only training on synthetic data. Through comprehensive ablations of pairwise, pointwise, and multitask J1 variants, we demonstrate the effectiveness of our approach across seed prompts, reward strategies, and training recipes. Qualitative analysis reveals that J1 develops systematic evaluation strategies, including dynamic criteria generation, reference answer creation, iterative self-correction of initial assessments, and feedback generation for low-quality responses.

## 1 Introduction

Better judgments can be made by learning how to reason, which is observed in both humans and machines. For models, the ability to judge predictions is a vital process that is applied at all stages of development: during training and inference to provide a reward or verification signal, and during final benchmark evaluation to judge performance. Classical evaluation using reward models typically outputs a score directly (Ouyang et al., 2022) without having an explicit reasoning step. Using pre-trained and aligned language models to act as judges instead, i.e., LLM-as-a-Judge, allows the possibility to generate chain-of-thought reasoning before making a decision, which was at first invoked by prompting (Zheng et al., 2023; Gu et al., 2024; Saha et al., 2024). Subsequently, iterative finetuning and direct preference optimization (DPO) methods were developed to improve these reasoning steps (Mahan et al., 2025; Wang et al., 2024c; Yu et al., 2025a; Saha et al., 2025). In this work, we investigate approaches for further improvements to judgment reasoning via online Reinforcement Learning (RL).

We introduce **J1** (Thinking-LLM-as-a-Judge via RL), a framework that incentivizes LLMs to think for evaluation through three primary aspects: (i) *Unified Verifiable Training*: we design a unified training recipe that converts all judgment tasks, from *both* verifiable (e.g., math problems) and typically subjective, non-verifiable prompts (e.g., user prompts from WildChat (Zhao et al., 2024)), into a format that can be optimized with RL from verifiable rewards. This allows us to train a single, generalist judge across diverse domains using only synthetic data. (ii) *Reasoning-Optimized Training*: we use GRPO (Shao et al., 2024) to directly optimize the quality of evaluation thoughts, in analogy to the approach in DeepSeek-R1 (Guo et al., 2025a). Guided by a seed prompt and targeted reward schemes, J1 teaches the model to reason critically about evaluation. (iii) *Multitask and Bias-Aware Judge*: we address positional bias through *consistency-based* rewards and, more importantly, develop

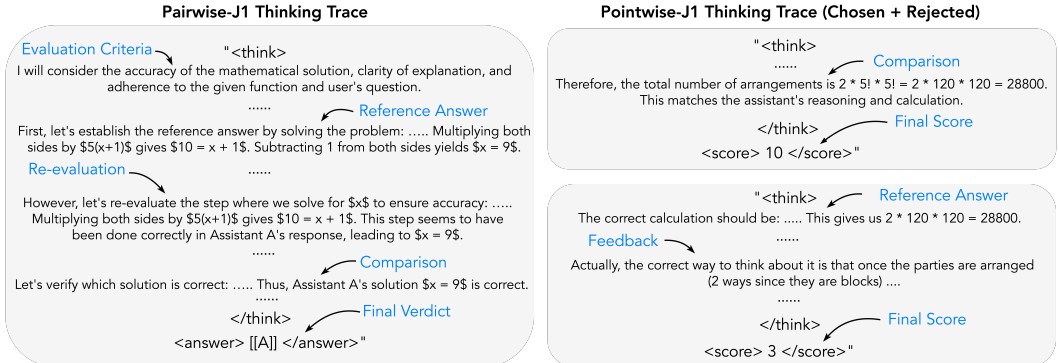

Figure 1: Illustration of the thinking patterns of pairwise and pointwise `J1` models during RL training. `J1` learns to outline evaluation criteria, generate reference answers, re-evaluate correctness, and compare between responses. Pairwise setup outputs a final verdict indicating the better response, pointwise generates a real-valued score, with a higher score for the better response. More examples of `J1`'s thinking traces are shown in Figures 6, 7, and 8.

a method to train inherently position-consistent *pointwise* judges using only pairwise supervision. We then unify these approaches into a single multitask model that is capable of performing both pointwise and pairwise evaluations.

After applying our best `J1` training recipe, we train judges on top of Llama-3.1-Instruct (Grattafiori et al., 2024) and Qwen3 (Yang et al., 2025) models at 8B, 32B, and 70B scales that achieve state-of-the-art performance across a variety of benchmarks: PPE (Frick et al., 2025), RewardBench (Lambert et al., 2025), JudgeBench (Tan et al., 2025), RM-Bench (Liu et al., 2025a), and FollowBenchEval (Saha et al., 2025). In particular, `J1` significantly outperforms: (i) models trained with SFT or Self-Taught reasoning (Wang et al., 2024c) and offline DPO (Saha et al., 2025); (ii) scalar RMs trained with a Bradley-Terry objective such as Skywork-RM (Shiwen et al., 2024); (iii) recent state-of-the-art generative reward models such as DeepSeek-GRM (Liu et al., 2025b) and Reasoning Reward Model (Guo et al., 2025b) that are trained on significantly more data; (iv) open-weight reasoning models like DeepSeek-R1-671B; and (v) closed reasoning models such as o1-mini (Jaech et al., 2024) and o3 (OpenAI, 2025). Crucially, `J1` achieves these results while only training a 32B model and leveraging a small amount of synthetic training data. Test-time scaling of `J1` via majority vote over multiple verdicts or averaging over multiple judgment scores leads to further improvements.

We provide detailed ablations and analysis of these results by comparing our best training strategy to other variants, that either modify the LLM-as-a-Judge setup (e.g., pairwise vs pointwise vs joint/multitasked, with vs without scores), bias mitigation strategies, reward modeling approaches, or seed thinking prompt. We show that our proposed *joint* model flexibly performs both pointwise and pairwise evaluations, while also being superior to both separately trained counterparts. In addition, we also analyze the distribution of the generated scores, thought lengths, and reasoning patterns within the thought generations. Our qualitative analysis shows that `J1` models learn to make better judgments by systematically outlining evaluation criteria, comparing responses against self-generated reference answers, critically re-evaluating their own initial assessments, and providing feedback (see Figures 1, 6, 7, and 8 for examples).

## 2 `J1`: THINKING-LLM-AS-A-JUDGE VIA REINFORCEMENT LEARNING

Our `J1` framework trains an LLM-as-a-Judge that generates chain-of-thought reasoning before scoring responses or making a preference judgment. Our primary setting is *pairwise* evaluation, where the judge takes an instruction $x$ and two responses $(a, b)$ to produce a verdict $y$ indicating the preferred response. The model's output consists of intermediate thought tokens $t$ followed by the final verdict $y$, conditioned on a seed prompt designed to elicit reasoning (see Appendix B).

This section details the three core components of our method. First, we describe the synthetic data generation process that creates verifiable tasks for RL training (§2.1). Next, we define the reward

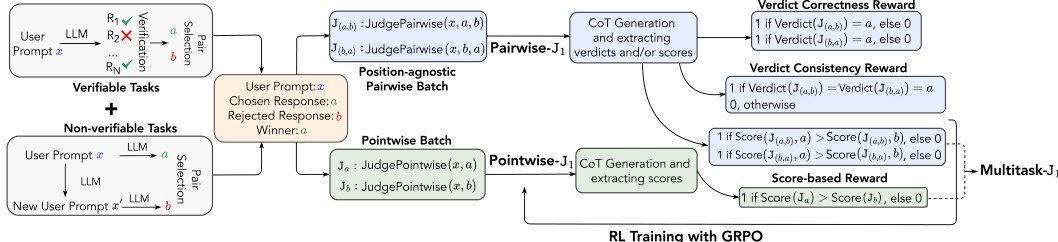

Figure 2: Reinforcement Learning recipes for training `J1` models. We generate synthetic preference pairs for both verifiable and non-verifiable tasks to create position-agnostic training batches. Rewards based on verdict correctness, consistency, and score alignment jointly optimize evaluation thoughts and verdicts using online GRPO. `Pointwise-J1` is trained *only* via distant supervision from pairwise labels. `MultiTask-J1` combines pairwise and pointwise formulation.

functions used to optimize for both judgment correctness and consistency (§2.2). Finally, we outline different judge formulations and training, including pairwise, pointwise, and multitask model (§2.3). An overview of the overall process is shown in Figure 2.

## 2.1 SYNTHETIC TRAINING DATA GENERATION

The goal of `J1` is to train a generalist judge for both verifiable and non-verifiable tasks. To achieve this, we follow the unified training dataset of synthetic preference pairs from Wang et al. (2024a), which removes the reliance on costly human annotations used in prior work (Liu et al., 2025b; Chen et al., 2025b). By building on the same data generation strategy from Saha et al. (2025), we can directly compare the effectiveness of our online RL approach against prior work using offline DPO on the same data distribution.

Our final 22K training data consists of 17K WildChat (Zhao et al., 2024) and 5K MATH (Hendrycks et al., 2021) prompts, along with their corresponding preference pairs. For WildChat, rejected responses are obtained by prompting an LLM to first generate a "noisy" variant of the original instruction and then produce a response to this noisy instruction (Wang et al., 2024c). See Figure 14 for the prompt and Table 5 for an example of a synthetically generated training pair. For MATH, rejected responses are sampled from generations by an LLM that do not lead to the gold answer. Given these preference pairs, we are thus able to convert the evaluation task into a verifiable task (i.e., predicting the better response), enabling the use of RL with verifiable rewards.

To address the known issue of *position bias* in pairwise judges (Wang et al., 2024b; Chen et al., 2024), we augment the training data by including both response orderings – $(x, a, b)$ and $(x, b, a)$, using the thinking seed prompt (Figure 9 in Appendix B). We construct training batches to be position-agnostic, i.e., both orderings of a given pair are processed in the same batch. As detailed in the next section, this batching strategy is crucial for implementing our consistency-based rewards.

## 2.2 REWARD MODELING

We adopt a straightforward and effective rule-based reward system designed to encourage accurate and consistent judgments.

**Verdict Correctness.** `J1`'s primary reward signal is binary. The model receives a reward of $+1$ if its final verdict correctly identifies the preferred response, and $0$ otherwise.

**Verdict Consistency.** To explicitly mitigate position bias, we introduce a consistency reward. A reward of $+1$ is granted only if the model produces the correct verdict for both input orderings of a response pair, i.e., for $(x, a, b)$ and $(x, b, a)$. An incorrect verdict on either ordering results in a reward of $0$.

We also explored adding format-based rewards to enforce the use of "<think>" tags, but found no noticeable performance benefit. The effects of different reward components are ablated in §4.2.

## 2.3 J1 FORMULATIONS AND TRAINING

Having described the training data and reward schemes, we now use GRPO (Shao et al., 2024) to jointly optimize thought generation and final judgments. We explore several training formulations that differ in their input/output formats and reward schemes.

**Pairwise J1 with Verdict (PaV).** Our first formulation, $\text{J1}_{\text{PaV}} : \text{prompt}_{\text{PaV}}(x, a, b) \to (t, y)$, receives a user question and a response pair, and generates thought tokens and the preferred response (as the final verdict). Figure 9 in Appendix B shows the seed thinking prompt. Figure 1 illustrates examples of judgment generation with this formulation.

**Pairwise J1 with Scores (PaS).** Instead of directly generating a verdict, our pairwise score-based variant $\text{J1}_{\text{PaS}} : \text{prompt}_{\text{PaS}}(x, a, b) \to (t, s_a, s_b)$, generates real-valued scores $s_a$, $s_b$ for response $a$ and $b$, respectively. The response that obtains the higher score is chosen as the final verdict. Figure 11 in Appendix B shows the corresponding thinking prompt. To train a model with this recipe, we replace the verdict-based reward with a score-based reward: a binary reward is assigned depending on whether the predicted scores are consistent with the gold verdict.

**Pairwise J1 with Scores&Verdict (PaVS).** In another pairwise variant $\text{J1}_{\text{PaVS}}$ : $\text{prompt}_{\text{PaVS}}(x, a, b) \to (t, s_a, s_b, y)$, the model generates scores for both responses as well as the final verdict, where the generated scores are interpreted as observations of unknown latent variables to help with the pairwise judgment task; therefore $y$ is directly used as the final verdict. Consequently, the reward is also computed using only the final verdict (and not the intermediate scores). Figure 12 in Appendix B shows the corresponding thinking prompt.

**Pointwise J1 (PoS).** We also introduce a pointwise judge, $\text{J1}_{\text{PoS}} : \text{prompt}_{\text{PoS}}(x, a) \to (t, s)$, which takes an instruction $x$ and a single response $a$ as input, and outputs a score $s$ that reflects the quality or reward of the response. Unlike pairwise judges, pointwise judges are inherently position-consistent. We train $\text{J1}_{\text{PoS}}$ via *distant supervision* with the same pairwise training data as used for all our pairwise variants. Each preference pair is split into two separate pointwise samples and the model is trained with a seed thinking prompt that assigns a score between 0 and 10 to a given response (see Figure 13 in Appendix B). Both scores are evaluated jointly, and the model receives a reward of 1 if the scores are consistent with the gold verdict. Since preference rankings are significantly easier to obtain than pointwise annotations, Pointwise-Thinking-LLM-Judge represents one of our novel contributions.

**Multitask Pairwise & Pointwise J1 (MT).** Finally, we unify the Pointwise and Pairwise (PaS) paradigms into a single multitask model by jointly training on both pairwise and pointwise data. This model has the advantage of being a unified model that can be evaluated in both setups and improves upon separately trained judges. As we show later, given the overall superiority of pairwise judgments over pointwise judgments, we evaluate the multitasked model in a pairwise setup for best results.

## 3 EXPERIMENTAL SETUP

**Training.** We implement J1 on top of verl (Sheng et al., 2025). All variants are trained on the 22K synthetic preference pairs described in Section 2.1. Appendix C includes other details of our experimental setup. Unless otherwise stated, all main J1 experiments are conducted with our primary Pairwise J1 with Verdict (PaV) recipe and the multitasked J1 model is evaluated in a pairwise setting.

**Evaluation.** We evaluate J1 on five pairwise judgment benchmarks, covering both verifiable and non-verifiable tasks, including multilingual instructions and responses from a wide range of LLMs.

- **Preference Proxy Evaluations (PPE)** (Frick et al., 2025) is a large-scale benchmark that links reward models to real-world human preference performance. It consists of two subsets: (i) **PPE Preference** (10.2K samples), human preference pairs from Chatbot Arena featuring 20 LLMs in 121+ languages, and (ii) **PPE Correctness** (12.7K samples), response pairs from four models across popular verifiable benchmarks (MMLU-Pro, MATH, GPQA, MBPP-Plus, IFEval). The first subset evaluates subjective preferences, while the second tests alignment in Best-of-N tasks.
- **JudgeBench** (Tan et al., 2025) (350 preference pairs) contains challenging response pairs that span knowledge, reasoning, math, and coding categories. Following Tan et al. (2025), we report results on the subset where the responses are generated by GPT-4o.

Table 1: Results on PPE Correctness comparing J1 against other LLM-Judges and reward models. All models are evaluated in a pairwise setup. †: Results from Liu et al. (2025b) and Frick et al. (2025). ‡: Trained solely on synthetically constructed preference pairs. Reported gains are relative to the corresponding base models.

| PPE Correctness | #Training Pref. Pairs | Overall | MMLU-Pro | MATH | GPQA | MBPP-Plus | IFEval |
|---|---|---|---|---|---|---|---|
| *Base LLM-as-a-Judge* | | | | | | | |
| Llama-3.1-8B-Instruct | – | 54.7 | 56.3 | 62.9 | 51.4 | 50.1 | 52.8 |
| Llama-3.3-70B-Instruct | – | 65.7 | 72.1 | 73.1 | 61.2 | 59.6 | 62.3 |
| Qwen3-32B (thinking) | – | 66.5 | 75.6 | 85.2 | 53.6 | 55.9 | 62.0 |
| *SOTA Scalar Reward Models* | | | | | | | |
| Armo-8B-v0.1† | 1000K | 61.2 | 66.0 | 71.0 | 57.0 | 54.0 | 58.0 |
| Skywork-Reward-Gemma-2-27B† | 80K | 54.7 | 55.0 | 46.2 | 44.7 | **69.1** | 58.3 |
| DeepSeek-BTRM-27B† | 237K | 66.7 | 68.8 | 73.2 | 56.8 | 68.8 | 66.0 |
| *SOTA Generative Reward Models* | | | | | | | |
| DeepSeek-GRM-27B† | 237K | 59.8 | 64.8 | 68.8 | 55.6 | 50.1 | 59.8 |
| EvalPlanner-Llama-8B | 22K‡ | 52.8 | 57.0 | 59.0 | 50.3 | 47.7 | 50.0 |
| EvalPlanner-Llama-70B | 22K‡ | 70.2 | 78.4 | 81.7 | 64.4 | 62.2 | 64.3 |
| *J1 Models (Ours)* | | | | | | | |
| J1-Llama-8B | 22K‡ | 59.2 $_{+4.5}$ | 65.6 $_{+9.3}$ | 70.0 $_{+7.1}$ | 53.2 $_{+1.8}$ | 53.1 $_{+3.0}$ | 54.0 $_{+1.2}$ |
| J1-Llama-70B | 22K‡ | 72.9 $_{+7.2}$ | 79.0 $_{+6.9}$ | 86.0 $_{+12.9}$ | 65.9 $_{+4.7}$ | 66.0 $_{+6.4}$ | 67.3 $_{+5.0}$ |
| J1-Qwen-32B | 22K‡ | 74.6 $_{+8.1}$ | 82.2 $_{+6.6}$ | 93.3 $_{+8.1}$ | 65.2 $_{+11.6}$ | 65.3 $_{+9.4}$ | 66.8 $_{+4.8}$ |
| J1-Qwen-32B-MultiTask | 22K‡ | **76.8** $_{+10.3}$ | **85.0** $_{+9.4}$ | **94.3** $_{+9.1}$ | **68.6** $_{+15.0}$ | 66.3 $_{+10.4}$ | **69.5** $_{+7.5}$ |

- **RM-Bench** (Liu et al., 2025a) (4K samples) assesses the robustness of reward models based on their sensitivity and resistance to subtle content differences and style biases.
- **FollowBenchEval** (Saha et al., 2025) (205 preference pairs) tests reward models for their ability to validate multi-level constraints in LLM responses (e.g., "Write a one sentence summary (less than 15 words) for the following dialogue. The summary must contain the word 'stuff'...").
- **RewardBench** (Lambert et al., 2025) (3K samples), similar to JudgeBench, consists of preference pairs from 4 categories of prompts: chat, chat-hard, safety, and reasoning.

Consistent with prior work, we report accuracy over a random ordering of paired responses for PPE, RewardBench, and RM-Bench. For JudgeBench and FollowBenchEval, we instead report *position-consistent accuracy*, where a sample is deemed correct only if the judge produces the correct verdict under both response orders. A more detailed analysis of position consistency is presented in Section 4.2. Model selection is based on overall accuracy on RewardBench. Inference is performed using vLLM (Kwon et al., 2023).

**Baselines.** We compare J1 to different categories of baselines: (i) non-thinking LLMs acting as judges in a zero-shot manner (e.g., Llama-3.3-70B-Instruct, GPT-4o (Hurst et al., 2024), (ii) thinking LLMs acting as judges (e.g., DeepSeek-R1-Distilled-Llama (Guo et al., 2025a), DeepSeek-R1, Qwen3-32B (Yang et al., 2025), OpenAI-o1-mini, o3 (OpenAI, 2025)) (iii) state-of-the-art scalar reward models (e.g., DeepSeek-BTRM-27B (Liu et al., 2025b), Armo (Wang et al., 2024a), Skywork-Reward-Gemma-2-27B (Shiwen et al., 2024)), (iv) state-of-the-art generative reward models that belong to the same category as J1 (e.g., EvalPlanner (Saha et al., 2025), DeepSeek-GRM-27B (Liu et al., 2025b)), and Reasoning Reward Model (Guo et al., 2025b). Additionally, for RewardBench, we compare J1 to all highest-ranked Generative Reward Models according to the leaderboard.[1]

# 4 RESULTS

## 4.1 MAIN RESULTS ON BENCHMARKS WITH Pairwise-J1 AND MultiTask-J1

**Results on PPE Correctness.** We first evaluate J1 on PPE Correctness because it directly tests RMs and LLM-Judges for their ability to improve popular reasoning benchmarks. This benchmark also offers a broader potential for improvement compared to others, such as RewardBench. Table 1 shows the results. J1-Qwen3-32B-MultiTask, our best J1 model, obtains state-of-the-art performance with an overall accuracy of 76.8, outperforming all previous methods by significantly large margins

---

[1] https://huggingface.co/spaces/allenai/reward-bench

Table 2: Results on five reward modeling benchmarks, where PPE includes both correctness and preference subsets of data. We compare J1 at different scales to EvalPlanner and general Thinking-LLMs (distilled-R1, o1-mini, o3, and R1). We report the default metric for each benchmark, where †: JudgeBench and FollowBenchEval use positional consistent accuracy. Reported gains are relative to the corresponding base models.

| Models | Overall | PPE | RewardBench | RM-Bench | JudgeBench† | FollowBenchEval† |
|---|---|---|---|---|---|---|
| Llama-3.1-8B-Instruct | 48.3 | 55.6 | 69.5 | 54.0 | 32.3 | 30.2 |
| DeepSeek-R1-Distilled-Llama-8B | 54.7 | 58.9 | 73.7 | 69.3 | 30.5 | 40.9 |
| EvalPlanner-Llama-8B | 56.2 | 54.3 | 83.0 | 68.1 | 30.2 | 45.3 |
| **J1-Llama-8B** | **61.9** $_{+13.6}$ | 59.8 $_{+4.2}$ | 85.7 $_{+16.2}$ | 73.4 $_{+19.4}$ | 42.0 $_{+9.7}$ | 48.3 $_{+18.1}$ |
| Llama-3.3-70B-Instruct | 64.3 | 65.8 | 85.4 | 69.5 | 48.6 | 52.2 |
| DeepSeek-R1-Distilled-Llama-70B | 67.4 | 68.6 | 86.9 | 80.8 | 46.0 | 54.6 |
| EvalPlanner-Llama-70B | 73.2 | 67.9 | 93.8 | 82.1 | 56.6 | 65.4 |
| **J1-Llama-70B** | **75.0** $_{+10.7}$ | 69.6 $_{+3.8}$ | 93.3 $_{+7.9}$ | 82.7 $_{+13.2}$ | 60.0 $_{+11.4}$ | 69.3 $_{+17.1}$ |
| Qwen3-32B | 77.3 | 66.5 | 90.9 | 88.1 | 70.8 | 70.0 |
| **J1-Qwen-32B-MultiTask** | **80.8** $_{+3.5}$ | 71.8 $_{+5.1}$ | 93.6 $_{+2.7}$ | 90.3 $_{+2.2}$ | 71.4 $_{+0.6}$ | 77.1 $_{+7.1}$ |
| OpenAI-o1-mini | 72.7 | 68.5 | 87.1 | 80.8 | 64.2 | 62.9 |
| OpenAI-o3 | 77.4 | 72.1 | 86.4 | 86.1 | 75.7 | 66.8 |
| DeepSeek-R1-671B | 78.4 | 72.3 | 90.6 | 88.6 | 68.9 | 71.7 |

Table 3: Results on PPE Correctness comparing judgment bias of pairwise and pointwise J1. *Consistent Accuracy*: proportion of examples judged correctly in both response orders. *Verdict Flip*: proportion of cases where the pairwise verdict changes when order is swapped. *Ties*: proportion of examples where the pointwise judge assigns equal scores.

| Models | Type | (a, b) Acc ↑ | (b, a) Acc ↑ | Consistent Acc ↑ | Verdict Flip/Ties ↓ |
|---|---|---|---|---|---|
| **J1-Llama-70B** | Pairwise | 72.9 | 72.3 | 61.2 | 21.9 |
| **J1-Llama-70B** | Pointwise | – | – | 65.0 | 13.7 |
| **J1-Qwen-32B** | Pairwise | 74.6 | 74.2 | 65.2 | 14.5 |
| **J1-Qwen-32B** | Pointwise | – | – | 69.3 | 13.0 |
| **J1-Qwen-32B-MultiTask** | Pairwise | 76.8 | 76.2 | 67.0 | 17.0 |
| **J1-Qwen-32B-MultiTask** | Pointwise | – | – | 70.6 | 10.5 |

($p < 0.0001$), including those trained on much more data (see column 2). Compared to related competing approaches that are generative reward models, J1-Qwen3-32B-MultiTask outperforms both EvalPlanner (Saha et al., 2025) by 6.8% (70.2 → 76.6), and DeepSeek-GRM-27B (Liu et al., 2025b) by 17% (59.8 → 76.8). J1 also improves upon the base Qwen3-32B model, a strong thinking-LLM, by a large 10.3% (66.5 → 76.8).

Furthermore, all J1 models at three different scales outperform their base counterparts, highlighting the generalizability of our recipe. In particular, this validates the effectiveness of J1's training methodology and use of online RL, compared to EvalPlanner, which is trained on the same data but with two iterations of DPO. Second, this also shows the effectiveness of J1's high-quality synthetic preference pairs, compared to the data used to train DeepSeek-GRM-27B. The latter is first fine-tuned on 1270K judge data, followed by stages of Reinforcement Learninng on 237K samples and further scaling at test time with a meta reward model across 32 generations. At a smaller scale, J1-Llama-8B is competitive with Armo-8B (scalar RM) and outperforms EvalPlanner-8B and a larger Skywork-Reward-Gemma-2-27B by significant margins (52.8 → 59.2 and 54.7 → 59.2, respectively).

**Results on RewardBench.** Table 6 in Appendix D shows a comparison of J1 with leading generative reward models on RewardBench. J1-Qwen-32B-MultiTask obtains an overall score 93.6, outperforming all previous generative reward models. Importantly, J1 is equally performant on all four categories of RewardBench. This suggests that J1 is a generalist judge that can be used for evaluating responses to both verifiable and non-verifiable prompt tasks, and in different stages of the LLM development process.

**Comparison of J1 with Thinking-LLMs.** Next, in Table 2, we compare J1 to several SOTA Thinking-LLMs on all benchmarks at three different scales (8B/32B/70B). These include DeepSeek-R1-Distill-Llama, DeepSeek-R1, Qwen3-32B, OpenAI-o1-mini, and OpenAI-o3. DeepSeek-R1-

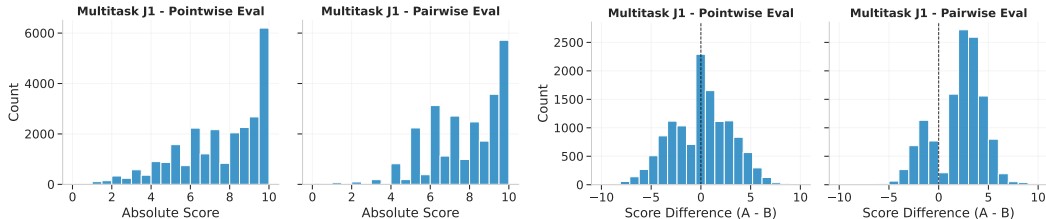

Figure 3: Distribution of Absolute Scores and $\Delta Score\ (Chosen - Rejected)$ on PPE Correctness from `J1-Qwen-32B-Multitask` when used in `pairwise` and `pointwise` settings. `Pairwise` exhibits sparser score distribution and larger score differences between Chosen and Rejected (ground-truth) responses. Note that all samples with a positive $\Delta$ in the range [0, 1) are correct predictions and count toward the '0' bar in the plot.

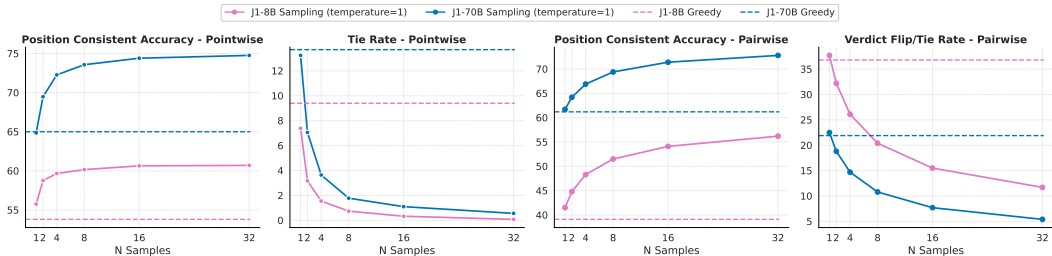

Figure 4: Test-time scaling of `J1` on PPE Correctness. We show greedy decoding results in dotted lines for comparison. As we sample more $N$, (i) position-consistent accuracy increases and (ii) tie rate decreases, for both pairwise and pointwise models, at both 8B and 70B scales.

Distilled-Llama is SFT-ed with 800K long CoTs from DeepSeek-R1 (a much larger 671B MoE model), starting from the same base models as `J1`. Thus, in a head-to-head comparison where the base models are the same, `J1` Llama models outperform DeepSeek-R1-Distilled-Llama across all benchmarks by large margins. Impressively, `J1-Qwen-32B-MultiTask` also outperforms state-of-the-art thinking models, R1 and o3, on three out of five benchmarks, while only training a 32B model with synthetic data. Through these comprehensive evaluations, `J1` thus reinforces the utility of online RL and synthetic data for training state-of-the-art Thinking-LLM-as-a-Judge models.

## 4.2 ABLATIONS AND ANALYSES OF `J1` IN POINTWISE VS. PAIRWISE SETUPS

In this section, we systematically analyze the impact of different `J1` formulations introduced in §2.3, focusing on following research questions: (i) How does pointwise setup compare to pairwise in mitigating position bias? (ii) What benefit does multitasking bring compared to separate `Pointwise-J1` and `Pairwise-J1` training? (iii) How can we design effective rewards for reinforcement learning in Thinking-LLM-as-a-Judge models? (iv) What is the impact of different thinking prompts on model behavior?

**Position Consistency.** There are two main ways of mitigating position bias in judgments – either by improving a pairwise judge itself (e.g., by training on both orders of data, adding consistency-based rewards, etc., like we do in `J1`) or by training a `Pointwise-J1` model that, by design, is position-consistent. While on one hand, pairwise judges can be position-inconsistent, pointwise judges on the other hand are consistent but lack the context of reference candidates to ground their evaluations on and hence are more prone to ties.

To understand this long-standing issue of judgment bias, in Table 3, we report (i) individual accuracy for both orders of responses, (ii) position-consistent accuracy, and (iii) verdict-flips/ties. We use "verdict flip" and "tie" to refer to the same metric which is the fraction of samples where either the verdict changes based on the response order (for pairwise judges) or both responses are scored equally (for pointwise judges). A *consistently correct* judge is thus expected to show higher position-consistent accuracy and lower verdict-flips/ties.

Table 4: Results of `Pairwise-J1` models trained with different reward schemes and seed prompts.

| Pairwise-J1 8B Variants | Overall | PPE | RewardBench | JudgeBench | RM-Bench | FollowBenchEval |
|---|---|---|---|---|---|---|
| *with Different Rewards* | | | | | | |
| Positive Reward for Correct Verdict | 61.8 | 59.8 | 85.7 | 42.0 | 73.4 | 48.3 |
| + Negative Reward for Incorrect Verdict | 60.4 | 59.6 | 85.4 | 44.9 | 70.8 | 42.0 |
| + Format Reward | 61.0 | 59.3 | 85.6 | 40.3 | 71.8 | 49.3 |
| *with Different Seed Prompts* | | | | | | |
| Thinking *(default - Figure 10)* | 61.8 | 59.8 | 85.7 | 42.0 | 73.4 | 48.3 |
| Plan & Execution *(Figure 9)* | 62.1 | 59.0 | 85.8 | 44.3 | 71.8 | 50.2 |

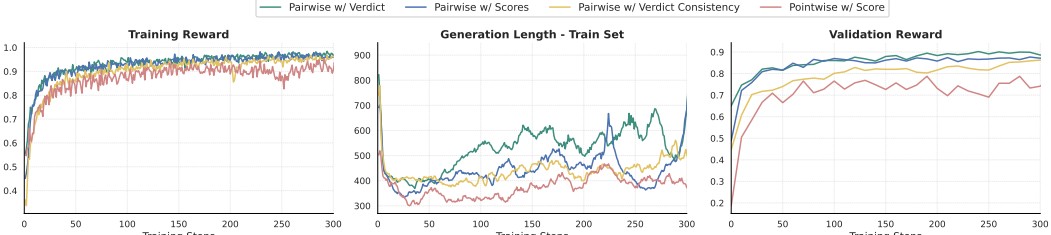

Figure 5: Reward and average generation length during training for different `J1-Llama-8B` models. `Pointwise-J1` is trained via *distant supervision* derived from pairwise preference data.

We observe that when judgment quality is measured by a stricter position-consistent accuracy, `Pointwise-J1` outperforms `Pairwise-J1`. However, if we consider a random ordering of response pairs, `Pairwise-J1` performs better. Moreover, the multitask formulation leverages the strengths of both pointwise and pairwise judges, outperforming separately trained judges in both random-order and position-consistency accuracy. In Table 8 and Appendix D, we also study the effect of training with both orders of data and verdict consistency rewards on `Pairwise-J1` models.

**Score Distribution of Pointwise vs Pairwise Evaluation.** Recall that `J1-Qwen-32B-MultiTask` has the advantage of being evaluated in both pointwise and pairwise settings. To understand the difference in scores assigned by the model in both these judgment settings, Figure 3 shows the distribution of (i) absolute scores and (ii) score differences between the chosen and rejected (ground-truth) responses. Both settings assign high scores more frequently (e.g., 8-10 are the tallest bars), but pairwise exhibits a sparser distribution and a larger gap between chosen and rejected responses than pointwise. This difference stems from their training objectives: pairwise directly contrasts both responses within the same input, enabling clearer differentiation reflected in the scores. In contrast, pointwise is trained with distant supervision on one response at a time, making fine-grained comparative judgments harder. Thus, combining both approaches can leverage their complementary strengths for improved quality.

**Effect of Test-time Scaling.** We explore test-time scaling of J1 by either conducting self-consistency over multiple verdicts or averaging over multiple pointwise scores. We show that these can lead to further improvements (see Table 12 and 13 in Appendix D). Here, we specifically analyze whether J1 models that generate scores can achieve more accurate judgments and fewer ties by averaging scores across multiple generations at test time. In Figure 4, we plot position-consistent accuracy and ties as a function of number of generations $N$. We observe improved position-consistent accuracy and reduction in ties for both Pairwise- and `Pointwise-J1` at both 8B and 70B scales. To the best of our knowledge, this is one of the first comprehensive analyses of Pointwise versus Pairwise judges, when both are trained using the same data.

**Effect of Reward Schemes and Seed Prompts for Training J1.** In Table 4, we first study the effect of different rewards for `Pairwise-J1` models. We obtain best results when only assigning positive rewards to correct verdicts – adding additional format rewards or negative rewards for incorrect verdicts marginally degrades performance. Next, we also analyze the effect of two different seed prompts that are used to elicit "thinking" in J1 models. Our default J1 *Thinking* prompt is motivated by DeepSeek-R1. Additionally, similar to EvalPlanner (Saha et al., 2025), we experiment with a prompt that instructs the model to first *plan* for the evaluation recipe, then *execute* the evaluation according to that recipe and the response(s), before generating the final verdict (Figure 10 in Appendix B). We find that J1 is robust to such choices, performing comparably with both prompts.

In fact, with a simpler Thinking prompt, the model tends to generate richer reasoning traces, including evaluation criteria, reference answers, re-evaluations, and detailed comparisons (see Figure 1).

**Reward and Thought Length Analysis of J1.** Figure 5 illustrates the training and validation reward, as well as average generation length throughout different stages of J1 training. As training progresses, the thought lengths of most pairwise judges converge at around 500 tokens, while the pointwise judge tends to generate shorter outputs, typically between 300 and 400 tokens due to the absence of *comparison*-style tokens. Training rewards for the pairwise variants exhibit a similar steady increase before converging. In Table 7 of Appendix D, we provide a comparison of different `Pairwise-J1` variants, all of which show comparable performance.

**Qualitative Analysis of J1's Thinking Traces.** In Figure 6, 7 and 8 we show three representative examples of J1's thinking traces. The first two are for verifiable math prompts and the third one is for a non-verifiable prompt. In Figure 6, we see that J1 identifies a calculation mistake in Assistant B's answer and provides critical feedback of how and where to improve the answer.

# 5 RELATED WORK

**Reward Models.** Reward Models have been instrumental in both training-time (Ouyang et al., 2022; Lambert et al., 2025) and test-time (Snell et al., 2025) alignment of LLMs. Traditional reward models are typically trained with the Bradley-Terry objective and output a scalar score indicating the reward of the response. This design frequently results in poor calibration and generalization across different prompts and responses (Sun et al., 2025; Zhang et al., 2025). Furthermore, such discriminative models do not fully leverage the generative capabilities of LLMs and therefore cannot be scaled up at test time, e.g., with long chain-of-thought or multiple generations (Wang et al., 2025; Shiwen et al., 2024). As a potential solution, generative reward models have emerged, which we discuss below.

**LLM-as-a-Judge and Generative Reward Models.** LLM-as-a-Judge and Generative Reward Models (GRMs) conceptually share a similar motivation – the language modeling head in LLMs is used to also output chain-of-thought (CoT) reasoning (in the form of critiques) before generating preference judgments or rewards (Kim et al., 2024a;b; Ankner et al., 2024; Mahan et al., 2025; Ye et al., 2025; Yu et al., 2025b; Zhang et al., 2025; Saha et al., 2025). Rewards in such models could either come from training a separate reward head (typically done in GRMs) or from the LM head itself generating real-valued scores as tokens (typically done in LLM-as-a-Judge). Prior work building LLM judges has mostly relied on either prompting (Zheng et al., 2023; Saha et al., 2024), rejection finetuning on self-generated CoTs (Wang et al., 2024c), or preference finetuning on CoT pairs using DPO (Mahan et al., 2025; Trivedi et al., 2024; Saha et al., 2025; Yu et al., 2025a).

Recently, in some concurrent studies, methods like DeepSeek-GRM (Liu et al., 2025b), JudgeLRM (Chen et al., 2025a), RM-R1 (Chen et al., 2025b), and Reward Reasoning Model (Guo et al., 2025b) use Reinforcement Learning for building reasoning judge models. We compare J1 to these methods (in Tables 1 and Table 4) and show that J1 achieves superior performance with significantly less data. This is achieved via J1's methodical novelty that span several axes. First, it is a training recipe that is based only on *synthetic* data. Second, it focuses on mitigating position bias (a long-standing issue in LLM-as-a-Judge development) and multitask learning for a single generalist judge, leading to the proposal of novel consistency rewards and `Pointwise-J1` models trained with *pairwise supervision only*. Consequently, we are able to comprehensively study different J1 variants that vary across sizes, modeling choices, seed prompts, and reward strategies, enabling us to draw important conclusions on building generalist thinking-judge models with state-of-the-art results.

**Reinforcement Learning with Verifiable Rewards.** J1 draws inspiration from the recent advancements in improving reasoning through Reinforcement Learning with verifiable rewards. Online optimization algorithms such as GRPO, when combined with accurate and robust rewards, have been shown to elicit enhanced *reasoning* in LLMs (Guo et al., 2025a; Team et al., 2025; OpenAI, 2025). In our approach, we construct preference pairs and assign verifiable rewards based on the correctness of the model's judgments. By optimizing over the generated thinking traces, J1 encourages LLMs to spend more test-time compute before deriving scores and judgments.

## 6 CONCLUSION

We proposed J1, an RL recipe for training Thinking-LLM-as-a-Judge models. Our key innovation lies in converting the judgment task into a verifiable one for all kinds of task prompts, both verifiable and non-verifiable, and then optimizing the thoughts and judgments using an online RL method. We trained J1 at 8B, 32B, and 70B scales, exploring pointwise, pairwise, and multitask formulations. Our generalist J1 judge models outperformed all baselines at their respective sizes, with J1-Qwen-32B also surpassing the much larger R1 and o3 models on some benchmarks. Furthermore, we find that with only pairwise supervision, we can train pointwise judges that effectively mitigate position bias. Finally, we proposed a multitask training strategy that combines the strengths of pairwise judging (comparative context) and pointwise judging (position consistency), achieving the best overall performance and highlighting the potential Thinking-LLM-as-a-Judge.

### REPRODUCIBILITY STATEMENT

All J1 models are built upon open-weight Llama and Qwen models. The prompts used for data generation and training of J1 models are provided in the Appendix. Furthermore, the training code for J1 is based on the open-source Verl repository, and detailed hyperparameters to facilitate experiment reproducibility are listed in Appendix C.

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

## A  J1'S THINKING TRACES

Here in Figure 6, Figure 7 and Figure 8 we show additional representative examples of J1's thinking traces. The first is for verifiable math prompts and the third one is for a non-verifiable prompt.

---

**Example 1 of J1's Thinking Trace for a Verifiable Math Question**

```
<think>

Okay, let's see.  The user wants to know which assistant's answer is better.  The
problem is about finding the sum of the sums of all two-element subsets of the set
{1,2,3,4,5,6}.

First, I need to check if both answers are correct.  The correct answer should be
105.  Let me verify that.  The number of two-element subsets in a 6-element set is
C(6,2) = 15, which matches the problem statement.  Each element appears in (6-1) =
5 subsets.  So each number from 1 to 6 is included 5 times in the total sum.  The
sum of the numbers 1 through 6 is 21.  So total sum would be 5*21 = 105.  That's
the correct answer.

Now, looking at Assistant A's answer.  They list all 15 subsets, compute each
sum correctly, add them up step by step, and get 105.  The final answer is boxed
correctly.  The steps are clear and methodical.

Assistant B's answer also lists the subsets and computes each sum.  However, when
adding them up, they make a mistake.  The sums they list are correct, but when
they add them, they group them incorrectly.  For example, they say 3 + 4 + 5
+ 6 + 7 + 5 + 6 + 7 + 8 + 7 + 8 + 9 + 9 + 10 + 11.  Let me add those numbers:
3+4=7, +5=12, +6=18, +7=25, +5=30, +6=36, +7=43, +8=51, +7=58, +8=66, +9=75, +9=84,
+10=94, +11=105.  Wait, that's 105.  But Assistant B says they simplified to 47.
That's a clear error.  They must have miscalculated when adding.  The final answer
they give is 47, which is wrong.

So Assistant A's answer is correct, while Assistant B's is incorrect.  The
user's problem requires the correct answer, so Assistant A's response is better.
Additionally, Assistant B's explanation of the addition step is flawed.  They
mention grouping the sums into 13 + 10 + 11 + 13 = 47, which doesn't make sense.
The correct total is 105.

Therefore, Assistant A's answer is accurate and follows the instructions properly,
while Assistant B's answer contains a critical error leading to an incorrect
result.  The user's instructions emphasized providing a clear, step-by-step
solution, which Assistant A did correctly.  Assistant B's answer, despite being
detailed, has a mistake in the final addition step, leading to an incorrect
answer.

</think>
```

Figure 6: Thinking trace of J1 for a verifiable prompt. J1 first self-generates a reference answer to the math problem (highlighted in brown). Subsequently, it checks the correctness of Assistant A's answer and judges it as correct (highlighted in green). Then it concludes that Assistant B's answer is incorrect and also provides feedback by pointing to a calculation mistake (highlighted in blue).



**Example 2 of J1's Thinking Trace for a Verifiable Math Question**

```
<think>

Okay, let's see.  The user is asking about the area of an isosceles right triangle
with hypotenuse endpoints at A(5,-5) and B(-1,-1).  Both assistants provided
answers, but I need to figure out which one is correct and better explained.

....So Assistant A's answer is correct.

Now Assistant B. They also calculated the hypotenuse as sqrt(52).  Then they set
up the equation for the legs using Pythagoras:  2x² = 52, leading to x²=26.  Then
they tried to compute the area as (1/2)*base*height, using hypotenuse as base and
x as height.  But wait, in a right triangle, the legs are the base and height.
If you use the hypotenuse as the base, the height isn't one of the legs.  The
correct height relative to the hypotenuse is (leg * leg)/hypotenuse, which is
(sqrt(26)*sqrt(26))/sqrt(52) = 26/sqrt(52) = sqrt(52)/2.  So when they multiplied
(sqrt(52)*sqrt(26))/2, that's (sqrt(52*26))/2.  Let's compute that:  52*26 = 1352.
sqrt(1352) is sqrt(52*26) = sqrt(4*13*26) = 2*sqrt(338) 2*18.38 = 36.76.  Divided
by 2 gives 18.38, which is not 26.  But Assistant B simplified it as (1/2)(52)(1)
= 26, which is incorrect.  They made a mistake in simplifying the product of
sqrt(52) and sqrt(26).  The correct product is sqrt(52*26) = sqrt(1352) = 36.76,
not 52.  So their final answer of 26 is wrong.

Therefore, Assistant A's answer is correct with 13, while Assistant B's is
incorrect.  The key error in B's solution is the incorrect simplification of
the product of the hypotenuse and the leg as 52 instead of the correct value.
Assistant A's approach is more straightforward and correct.  Both solutions are
step-by-step, but B's mistake in the final calculation leads to an incorrect
answer.  So Assistant A's response is better.

</think>
```



Figure 7: Thinking trace of J1 for a verifiable math prompt. The model analyzes Assistant B's answer in a step-by-step manner, finds an error in the simplification step, and provides detailed feedback (highlighted in blue).

---

**Example 3 of J1's Thinking Trace for a Non-Verifiable Writing Prompt**

```
<think>

Okay, let's see.  The user is asking about the requirements for arc flash PPE,
specifically mentioning maintenance, use, inspections, and shock protection areas
like limited and restricted approach boundaries.  They want the answer to be for
only qualified persons.

Looking at Assistant A's answer:  They talk about qualified persons needing to
identify PPE, ensure maintenance and inspections, understand shock protection
areas, and train others.  That covers the points mentioned.  But wait, the user's
question is about the requirements for arc flash PPE, so maybe Assistant A is
adding more about training others, which might be relevant but not directly the
requirements for the PPE itself.

Assistant B's answer starts by stating only qualified persons can work on exposed
parts.  They mention being trained, knowing equipment, hazards, PPE, inspections,
and understanding the approach boundaries.  This seems more focused on the
qualifications and responsibilities of the person, which aligns with the user's
instruction to make the answer for only qualified persons.

The user's question includes the requirements for PPE and shock protection areas.
Both answers mention these, but Assistant B's answer is more concise and directly
addresses the requirements without adding extra info about training others.  Also,
Assistant B starts by stating that only qualified persons are allowed, which
directly answers the user's instruction.  Assistant A's answer might be more
detailed but includes elements not directly asked for.  So Assistant B's answer
is more precise and better follows the user's specific request.

</think>
```

Figure 8: Thinking trace of J1 for a non-verifiable prompt.

## B    PROMPT TEMPLATES

Figure 9 shows the seed prompt to train our primary J1 recipe: `Pairwise-J1` with verdict. Figure 10 shows an alternative seed prompt for training a similar `Pairwise-J1` with Verdict setup. Motivated by EvalPlanner (Saha et al., 2025), this prompt instructs the model to first *plan* the evaluation recipe and then *execute* it as part of the thinking process. Figure 11 and Figure 12 show our prompts for "`Pairwise-J1` with Scores" and "`Pairwise-J1` with Scores&Verdict" variants respectively. Finally, we adapt our pairwise prompts to a pointwise prompt used to train our `Pointwise-J1` model that instructs the model to think and assign real-valued scores between 0 and 10, shown in Figure 13.

---

**Thinking Seed Prompt Template for Training Pairwise J1 with Verdict**

```
You are given a user question and two responses from two AI assistants.  Your task
is to act as an impartial judge and evaluate which response better follows the
user's instructions and provides a higher-quality answer.

First, provide your reasoning within <think> and </think> tags.  This should
include your evaluation criteria for a high-quality response, a detailed
comparison of the two responses, and when helpful, a reference answer as part of
your evaluation.  Be explicit in your thought process, referencing your criteria
and explaining how each response aligns with or deviates from them.

Avoid any position biases and ensure that the order in which the responses
were presented does not influence your decision.  Do not allow the length of
the responses to influence your evaluation.  Do not favor certain names of the
assistants.  Be as objective as possible.

Finally, provide your verdict within <answer> and </answer> tags, strictly
following this format:
- <answer> [[A]] </answer> if Assistant A is better
- <answer> [[B]] </answer> if Assistant B is better

Below are the user's question and the two responses:

[User Question]
{instruction}

[The Start of Assistant A's Answer]
{response A}
[The End of Assistant A's Answer]

[The Start of Assistant B's Answer]
{response B}
[The End of Assistant B's Answer]
```

Figure 9: Thinking seed prompt template for `Pairwise-J1` with Verdict.

EvalPlanner-style Seed Prompt Template for Training Pairwise J1 with Verdict

```
You are given a user question and two responses from AI assistants.  Your task
is to act as an impartial judge and determine which response better follows the
user's instructions and provides a higher-quality answer.

First, build an evaluation plan that can then be executed to assess the response
quality.  Whenever appropriate, you can choose to also include a step-by-step
reference answer as part of the evaluation plan.  Enclose your evaluation plan
between the tags <plan> and </plan>.

Next, execute the plan step-by-step to evaluate the two responses.  Avoid copying
the plan when doing the evaluation.  Please also only stick to the generated
plan and provide an explanation of how the plan is executed to compare the two
responses.  Avoid any position biases and ensure that the order in which the
responses were presented does not influence your decision.  Do not allow the
length of the responses to influence your evaluation.  Do not favor certain names
of the assistants.  Be as objective as possible.  Enclose your plan execution
between the tags <execution> and </execution>.

Finally, output your final verdict by strictly following this format:
- <answer> [[A]] </answer> if Assistant A is better
- <answer> [[B]] </answer> if Assistant B is better

Below are the user's question and the two responses:

[User Question]
{instruction}

[The Start of Assistant A's Answer]
{response A}
[The End of Assistant A's Answer]

[The Start of Assistant B's Answer]
{response B}
[The End of Assistant B's Answer]
```

Figure 10: EvalPlanner-style (plan + execution) Prompt template for `Pairwise-J1` with Verdict.

```
Thinking Seed Prompt Template for Training Pairwise-J1 with Scores

You are given a user question and two responses from two AI assistants.  Your task
is to act as an impartial judge and evaluate which response better follows the
user's instructions and provides a higher-quality answer.

First, provide your reasoning within <think> and </think> tags.  This should
include your evaluation criteria for a high-quality response, a detailed
comparison of the two responses, and when helpful, a reference answer as part of
your evaluation.  Be explicit in your thought process, referencing your criteria
and explaining how each response aligns with or deviates from them.

Avoid any position biases and ensure that the order in which the responses
were presented does not influence your decision.  Do not allow the length of
the responses to influence your evaluation.  Do not favor certain names of the
assistants.  Be as objective as possible.

Finally, assign the assistant's response a score from 0 to 10, using either an
integer or a decimal with up to 0.1 precision, with a higher score indicating a
higher-quality response that better satisfies the criteria.  Enclose the scores
within the tags <score_A> </score_A>, and <score_B> </score_B>.

Format your output like this:
<think> your_thinking_process </think>
<score_A> your_score_a </score_A> <score_B> your_score_b </score_B>

Below are the user's question and the two responses:

[User Question]
{instruction}

[The Start of Assistant A's Answer]
{response A}
[The End of Assistant A's Answer]

[The Start of Assistant B's Answer]
{response B}
[The End of Assistant B's Answer]
```

Figure 11: Thinking seed prompt template for training `Pairwise-J1` with Scores.

Thinking Seed Prompt Template for Training Pairwise J1 with Verdict and Score

```
You are given a user question and two responses from two AI assistants.  Your task
is to act as an impartial judge and evaluate which response better follows the
user's instructions and provides a higher-quality answer.

First, provide your reasoning within <think> and </think> tags.  This should
include your evaluation criteria for a high-quality response, a detailed
comparison of the two responses, and when helpful, a reference answer as part of
your evaluation.  Be explicit in your thought process, referencing your criteria
and explaining how each response aligns with or deviates from them.

Avoid any position biases and ensure that the order in which the responses
were presented does not influence your decision.  Do not allow the length of
the responses to influence your evaluation.  Do not favor certain names of the
assistants.  Be as objective as possible.

Finally, assign the assistant's response a score from 0 to 10, using either an
integer or a decimal with up to 0.1 precision, with a higher score indicating a
higher-quality response that better satisfies the criteria.  Enclose the scores
within the tags <score_A> </score_A>, and <score_B> </score_B>.

Finally, provide your verdict within <answer> and </answer> tags, strictly
following this format:
- <answer> [[A]] </answer> if Assistant A is better
- <answer> [[B]] </answer> if Assistant B is better

Below are the user's question and the two responses:

[User Question]
{instruction}

[The Start of Assistant A's Answer]
{response A}
[The End of Assistant A's Answer]

[The Start of Assistant B's Answer]
{response B}
[The End of Assistant B's Answer]
```

Figure 12: Thinking seed prompt template for training `Pairwise-J1` with Verdict and Score.

---

**Thinking Seed Prompt Template for Training Pointwise J1**

```
You are given a user question and a response from an AI assistant.  Your task
is to act as an impartial judge and evaluate how well the response fulfills the
user's instructions.  You will be shown multiple responses to the same prompt, but
only one at a time.  Evaluate each response independently.

Think carefully about how to assess the quality of the response, and enclose your
reasoning within <think> and </think> tags.  Your reasoning should include your
evaluation criteria, a clear understanding of what an ideal response would look
like for this particular question, and a concrete example of such an ideal or
reference answer if possible.  Then compare the assistant's response to your
ideal or reference answer, explaining how it aligns with or deviates from your
expectations.  Be specific and avoid vague or overly general judgments.  Remain as
objective as possible.

Finally, assign the assistant's response a score from 0 to 10, using either an
integer or a decimal with up to 0.1 precision.  A higher score should indicate a
higher-quality response.  Enclose the score within <score> and </score> tags.

Format your output like this:
<think> your_thinking_process </think>
<score> your_score </score>

Below are the user's question and the assistant's response:

[User Question]
{instruction}

[The Start of the Assistant's Answer]
{response}
[The End of the Assistant's Answer]
```

Figure 13: Thinking seed prompt template for training `Pointwise-J1`.

---

**Prompt for Creating Preference Pairs for J1 Training**

```
Below is a conversation between an user and an AI Assistant.
{instruction}

The start of Assistant's Answer
{baseline response}

Please first generate a modified instruction that is highly relevant but not
semantically identical to the instruction above from the user.  Then write a
high-quality answer which is a good response to the modified instruction but not
a good response to the original user question.  IMPORTANT: Please strictly follow
the following format:

User Question Modified

<provide a modified instruction here>

The start of Assistant's answer to the modified instruction

<provide a high-quality response to the modified instruction>

The end of Assistant's answer to the modified instruction
```

Figure 14: Given a prompt and a baseline response, the prompt asks the LLM to generate a noisy instruction and then a response to that noisy instruction. The response to the noisy instruction subsequently is selected as the rejected response to the original instruction.

Table 5: An example of a synthetically generated training pair for J1 training. The rejected response is a good response to the noisy instruction, thus making it a bad response to the original instruction.

| Original Question | Noisy Question | Chosen Response | Rejected Response |
|---|---|---|---|
| There are 3 objects arranged in a line. Someone says "the table is to the right of the bag, the table is to the left of the box and the bag is to the right of the box." Is the situation just described possible or impossible? Before answering, take a deep breath. Think step by step before giving a final answer. My survival depends on your answer being correct, so concentrate hard.Please analyze this new situation and determine whether it is possible or impossible. | There are 3 objects arranged in a line. Someone says "the table is to the right of the bag, the box is to the left of the table and the bag is to the left of the box." Please analyze this new situation and determine whether it is possible or impossible. | Let's break down the statements: "The table is to the right of the bag." "The table is to the left of the box." "The bag is to the right of the box." From statement 3, we know the order is: box -> bag From statement 1, we know the order is: bag -> table Combining these two, we get: box -> bag -> table Now, let's look at statement 2: "The table is to the left of the box." This statement contradicts our previous conclusion. If the table is to the left of the box, then the correct order should be: table -> box -> bag However, this contradicts statement 1, which says the table is to the right of the bag. Therefore, I conclude that the situation described is IMPOSSIBLE. The statements are mutually contradictory, and it's not possible for all three statements to be true at the same time. " | Let's break down the given statements: The table is to the right of the bag. The box is to the left of the table. The bag is to the left of the box. From statement 3, we know the order is: bag -> box From statement 2, we know the order is: box -> table Now, let's combine these orders: bag -> box -> table (from statements 2 and 3) Statement 1 says the table is to the right of the bag, which is consistent with this order. Conclusion Based on the analysis, it is POSSIBLE for all three statements to be true at the same time. The correct order is: bag -> box -> table. |

## C   EXPERIMENTAL SETUP

For training, the policy actor generates 5 rollouts per prompt using ancestral sampling with temperature $1.0$. Training regime uses a learning rate of $1e-6$ (decayed to $3e-7$ in later steps for pairwise `J1-Llama-70B`), and a total batch size of 512. The maximum sequence length is set to 4096 tokens for both inputs and outputs.

We experimented with different KL coefficients from $\{0.1, 0.01, 0.001, 0\}$ for `J1-Llama-8B`, and selected $0.01$ as the best-performing value based on development set accuracy. For `J1-Llama-70B`, we set the KL coefficient to 0 to encourage more exploration. Preliminarily experiment with entropy bonus during training showed degraded performance: the model tends to generate longer but more repetitive output. See Table 10 in Appendix D for comparison of KL penalty and entropy bonus.

We use $8\times$A100 to train `J1-Llama-8B`, $32\times$A100 GPUs for `J1-Qwen-32B` and $64\times$A100 GPUs for `J1-Llama-70B`. Inference is done using $8\times$A100 GPUs. Tensor parallelism is set to 8 for both training and inference. During inference, we maintain a maximum generation length of 4096 tokens. For inference-time scaling we use sampling with top-p of $0.95$ and temperature of 1 for Llama, and the default $0.6$ for Qwen3.

## D   ADDITIONAL RESULTS

**Comparison of Different `Pairwise-J1` Models.** In Table 7, we compare the three `Pairwise-J1` variants, that as part of the final answer, generate either: (i) only the final verdict, (ii) only real-valued scores for both responses, or (iii) both. We observe that predicting the verdict (without the scores) performs as well as other variants. Having access to scores, however, has other advantages e.g., in quantifying the degree of preference or to rank across multiple responses. Next, in Table 8, we compare the effect of training `Pairwise-J1` models with both orders of data and with verdict consistency rewards. We observe that adding a consistency reward improves position-consistency accuracy of `J1` models and consequently, also reduces the fraction of verdict flips.

**Comparison of Different Decoding Hyperparameters for J1 Models.** In Table 9, we compare greedy decoding to temperature sampling of `J1` models. We find that our models are robust to such choices, exhibiting consistent performance with negligible variance.

Table 6: Results on RewardBench comparing our `J1-Qwen-32B-MultiTask` model with other top performing generative reward models. Results for most of these models are from the RewardBench leaderboard.

| Models | Overall | Chat | Chat-Hard | Safety | Reasoning |
|---|---|---|---|---|---|
| *Open and Closed LLM-as-a-Judge* | | | | | |
| Llama3.1-8B-Instruct | 69.5 | 92.7 | 46.1 | 64.4 | 74.7 |
| Llama3.3-70B-Instruct | 85.4 | 96.9 | 77.4 | 77.6 | 89.6 |
| Llama3.1-405B-Instruct | 84.1 | 97.2 | 74.6 | 77.6 | 87.1 |
| Claude-3.5-Sonnet | 84.2 | 96.4 | 74.0 | 81.6 | 84.7 |
| GPT-4o | 86.7 | 96.1 | 76.1 | 88.1 | 86.6 |
| Gemini-1.5-Pro-0514 | 88.2 | 92.3 | 80.6 | 87.9 | 92.0 |
| OpenAI-o1-mini | 87.1 | 94.4 | 78.7 | 80.9 | 94.2 |
| OpenAI-o3 | 86.4 | 92.7 | 80.5 | 79.8 | 92.7 |
| DeepSeek-R1 | 90.6 | 95.3 | 83.6 | 86.4 | 97.4 |
| *SOTA Generative Reward Models* | | | | | |
| CompassJudger CJ-1-32B (Cao et al., 2024) | 85.4 | 97.8 | 65.6 | 86.1 | 92.2 |
| facebook/Self-taught-evaluator-llama3.1-70B (Wang et al., 2024c) | 90.0 | 96.9 | 85.1 | 89.6 | 88.4 |
| Salesforce/SFR-nemo-12B-Judge-r (Wang et al., 2025) | 90.3 | 97.2 | 82.2 | 86.5 | 95.1 |
| SF-Foundation/TextEval-OffsetBias-12B | 91.0 | 91.9 | 86.6 | 92.0 | 93.6 |
| Reward Reasoning Model (Guo et al., 2025b) | 91.2 | 94.7 | 81.1 | 90.7 | 98.3 |
| RM-R1 (Chen et al., 2025b) | 91.4 | 95.3 | 83.1 | 91.9 | 95.2 |
| AtlaAI/Selene-1 (Alexandru et al., 2025) | 92.4 | 97.8 | 84.0 | 92.2 | 95.7 |
| R-I-S-E/RISE-Judge-Qwen2.5-32B (Yu et al., 2025a) | 92.7 | 96.6 | 83.3 | 91.9 | 98.8 |
| Salesforce/SFR-LLaMa-3.1-70B-Judge-r (Wang et al., 2025) | 92.7 | 96.9 | 84.8 | 91.6 | 97.6 |
| Skywork/Skywork-Critic-Llama-3.1-70B (Shiwen et al., 2024) | 93.3 | 96.6 | 87.9 | 93.1 | 95.5 |
| SF-Foundation/TextEval-Llama3.1-70B | 93.5 | 94.1 | 90.1 | 93.2 | 96.4 |
| **J1-Qwen-32B-MultiTask** (Ours) | **93.6** | 96.4 | 89.5 | 90.5 | 98.1 |

Table 7: Results of `Pairwise-J1` models trained with different recipes. $x$ : input instruction, $a, b$ : pair of responses, $t$ : intermediate thought, $y$: verdict, $s_a, s_b$: scores.

| Pairwise-J1 8B Variants | Overall | PPE | RewardBench | JudgeBench | RM-Bench | FollowBenchEval |
|---|---|---|---|---|---|---|
| w/ Verdict : $(x, a, b) \rightarrow (t, y)$ | 63.9 | 59.8 | 85.7 | 42.0 | 73.4 | 48.3 |
| w/ Scores: $(x, a, b) \rightarrow (t, s_a, s_b)$ | 63.4 | 60.2 | 85.8 | 41.7 | 72.3 | 46.3 |
| w/ Scores&Verdict: $(x, a, b) \rightarrow (t, s_a, s_b, y)$ | 61.7 | 59.5 | 85.1 | 41.4 | 71.5 | 41.0 |

**Effect of KL Penalty and Entropy Bonus in GRPO for training J1.** In Table 10, we study the effect of KL Penalty and Entropy Bonus in GRPO when training a Pairwise `J1-Llama-8B` model. In our experiments, we find that more exploration generally leads to some degradation in performance.

**Comparison of J1 with a Scalar RM Trained on Same Data.** In Table 11, we compare `J1` to a scalar RM trained on the same base model (Llama-3.1-8B-Instruct) using the same training data. We observe that `J1` outperforms the corresponding scalar model on four out of five benchmarks with the maximum improvement coming in the hardest RM-Bench benchmark.

Table 8: Results on PPE Correctness and JudgeBench comparing different position-bias mitigation strategies for `Pairwise-J1`. *Consistent Accuracy*: examples are judged correctly in both response orders. *Verdict Flip*: cases where the pairwise verdict changes when response order is swapped.

| `Pairwise-J1 8B Variants` | PPE Correctness | | | | JudgeBench | | | |
|---|---|---|---|---|---|---|---|---|
| | (a, b) Acc ↑ | (b, a) Acc ↑ | Consistent Acc ↑ | Verdict-Flip ↓ | (a, b) Acc ↑ | (b, a) Acc ↑ | Consistent Acc ↑ | Verdict-Flip/Ties ↓ |
| Llama-3.1-8B-Instruct | 54.7 | 54.1 | 30.2 | 44.1 | 67.4 | 42.3 | 32.3 | 37.4 |
| Random Single-order Data | 58.3 | 57.6 | 38.3 | 36.7 | 48.3 | 59.4 | 36.6 | 32.9 |
| Both-order data | 59.2 | 58.4 | 39.1 | 36.8 | 63.1 | 51.4 | 42.0 | 27.7 |
| Verdict Consistency Reward | 58.4 | 58.2 | 43.9 | 28.7 | 52.3 | 64.6 | 45.4 | 26.0 |

Table 9: Comparison of Greedy Decoding and Temperature Sampling showing the robustness of `J1` models to different decoding temperatures.

| Model | Decoding Temperature | PPE Correctness |
|---|---|---|
| `J1-Llama-70B` | Greedy (with t=0.0) | 72.9 |
| `J1-Llama-70B` | Temperature Sampling (8 seeds with t=0.6) | $72.8 \pm 0.1$ |
| `J1-Qwen-32B-MultiTask` | Greedy (with t=0.0) | 76.8 |
| `J1-Qwen-32B-MultiTask` | Temperature Sampling (8 seeds with t=0.6) | $77.0 \pm 0.1$ |

Table 10: Ablation studies on `Pairwise-J1` with verdict with KL penalty and entropy bonus.

| `Pairwise-J1 8B Variants` | Overall | PPE | RewardBench | JudgeBench | RM-Bench | FollowBenchEval |
|---|---|---|---|---|---|---|
| w/ KL Penalty | 63.9 | 59.8 | 85.7 | 42.0 | 73.4 | 48.3 |
| w/o KL Penalty | 61.1 | 59.6 | 84.9 | 42.9 | 69.8 | 48.3 |
| w/o Entropy Bonus | 63.9 | 59.8 | 85.7 | 42.0 | 73.4 | 48.3 |
| w/ Entropy Bonus | 58.3 | 59.1 | 84.1 | 39.4 | 71.7 | 42.4 |

Table 11: Comparison of `J1` with a scalar RM trained on same data.

| Model | Overall | PPE | RewardBench | JudgeBench | RM-Bench | FollowBenchEval |
|---|---|---|---|---|---|---|
| Bradley-Terry (scalar) | 66.6 | 57.5 | 82.6 | 74.0 | 51.1 | 67.8 |
| `J1-Llama-8B` (generative) | 69.2 | 59.8 | 85.7 | 73.4 | 58.5 | 68.8 |

Table 12: Test-time scaling of `Pointwise-J1` models on PPE Correctness. Judgments are made based on the average scores of the responses.

| `Pointwise-J1` | Overall | MMLU-Pro | MATH | GPQA | MBPP-Plus | IFEval |
|---|---|---|---|---|---|---|
| **J1-Llama-70B** - *Greedy* | 65.0 | 73.4 | 77.4 | 58.9 | 60.7 | 54.6 |
| **J1-Llama-70B** - *Sampling* | 64.9 | 74.0 | 79.7 | 58.2 | 55.7 | 59.7 |
| **J1-Llama-70B (Mean-Score@32)** | 74.8 | 81.2 | 87.6 | 67.3 | 81.9 | 70.8 |

Table 13: Test-time scaling of `Pairwise-J1` models on PPE Correctness. Judgments are made based on majority vote over multiple verdicts.

| `Pairwise-J1` | Overall | MMLU-Pro | MATH | GPQA | MBPP-Plus | IFEval |
|---|---|---|---|---|---|---|
| **J1-Llama-70B** - *Greedy* | 72.9 | 79.0 | 86.0 | 65.9 | 66.0 | 67.3 |
| **J1-Llama-70B (SC@32)** | 73.7 | 79.9 | 88.1 | 66.5 | 66.5 | 67.2 |

