# OpenReview forum: "J1: Incentivizing Thinking in LLM-as-a-Judge via Reinforcement Learning"
_ICLR.cc/2026/Conference — ICLR 2026 Poster_

### Official Review · Reviewer_TdB2 · 2025-10-31

**Soundness:** 3
**Presentation:** 3
**Contribution:** 3
**Rating:** 6
**Confidence:** 4

**Summary:**

This paper introduces an RL-based method for teaching an LLM-as-a-Judge to think before delivering a verdict/score. The authors investigate and compare various pairwise and pointwise strategies, and conduct meaningful analysis and ablations over these settings

**Strengths:**

1. The writing is clear and easy to follow.
2. The evaluation is comprehensive, spanning many relevant datasets and baselines.
3. The approach, particularly the formulation in Section 3.2, is clear and well-motivated. Many meaningful ablations over the approach are provided.

**Weaknesses:**

1. There is some disconnect between Sections 2.3 and Section 4.1. For example, it's not clear which formulations from Section 2.3 were used to train the models evaluated in Section 4.1. Section 3 could be used to provide more information about the trained models.
2. The role of the consistency reward is unclear. From Figure 5, it appears to slightly underperform the other pairwise settings, but a more thorough analysis (e.g., as a setting in Table 4) is needed.
3. In real-world settings, automatic evaluations often span multiple dimensions (e.g., helpfulness, factuality, safety). How could the J1 framework be adapted for this multi-task setting?
4. A couple of statements could use some qualification. For example, in L.180, the authors claim that "pointwise judges are inherently consistent", which is true in reference to positional consistency (because pointwise judges have no notion of position), but more broadly incorrect (e.g., sampling from the same pointwise judge is not guaranteed to produce consistent results). Similarly, in L.311, it states that J1-Qwen-32B-MultiTask outperforms "all previous methods" on RewardBench, but this is only true against other generative approaches; a number of classified approaches outscore J1-Qwen-32B-MultiTask [1].

[1] https://huggingface.co/spaces/allenai/reward-bench

**Questions:**

See weaknesses. Also,
1. [2] is recent work that trains a pointwise judge with RL, but they observe that response length explodes to over 10k tokens during training (similar to DeepSeek-R1-Zero [3]), whereas you observe that it initially falls and settles around 500 tokens. Why is this? Would performance be better if the models reasoned for longer before providing a score/verdict?
2. In L.262, the authors state the results are statistically significant "(p<0.0001)". How was this statistical testing/resampling done?

[1] https://huggingface.co/spaces/allenai/reward-bench

[2] https://arxiv.org/pdf/2504.10337

[3] https://arxiv.org/pdf/2501.12948

---

> ### Author Response · Authors · 2025-11-20
>
> We thank the reviewer for the feedback and for appreciating our comprehensive evaluation, well-motivated approach, and insightful ablations. We address the questions below:
>
> **1. Model Formulations (Section 2.3 vs 4.1)**
> > which formulations from Section 2.3 were used to train the models evaluated in Section 4.1? Provide more information about the trained models in Section 3
>
> We thank the reviewer for this suggestion. While Section 2.3 details our specific training formulations (Pairwise with Verdict, Pairwise with Scores, Multitask, etc.), we agree that the mapping to the specific model names in Section 4.1 could be more explicit.
>
> To clarify:
> - J1-Llama-8B/70B: These models utilize our primary recipe, Pairwise-J1 with Verdict. This is the default formulation corresponding to the seed prompt in Figure 9.
> - J1-Qwen-32B-MultiTask: we explicitly noted in lines 190-191 (original submission) that multitasked model is primarily evaluated in a pairwise setup for best results
>
> In the revised draft uploaded, we have added a line in Section 3 (in blue) to specifically clarify this.
>
> **2. Role of the Consistency Reward**
>
> We point the reviewer to Table 8 in the Appendix (discussed in lines 373-375), which presents a direct ablation study of the consistency reward. Comparing the “Both-order data” and “ Verdict Consistency Reward” rows, we directly ablate the weight of 0 vs. 1 for the consistency term. On both PPE and JudgeBench we observe significant improvements in Consistent Accuracy (+4.8 for PPE and +3.4 for JudgeBench) and a corresponding drop in the verdict flip ratio. This shows that the model training is highly responsive to the consistency reward signal.
>
> **3. Multi-dimensional Evaluation**
> > In real-world settings, automatic evaluations often span multiple dimensions (e.g., helpfulness, factuality, safety). How could the J1 framework be adapted for this multi-task setting?
>
> The J1 framework is naturally adaptable to these settings as our current evaluation benchmarks already cover these dimensions. For example:
> - RewardBench: Explicitly evaluates "Safety," "Chat" (Helpfulness), and "Reasoning" categories. As shown in Table 6, J1-Qwen-32B-MultiTask achieves strong performance across all these distinct subsets.
> - PPE: The Correctness subset evaluates specific domains like coding (MBPP-Plus) and math, while the Preference subset (10.2k pairs) directly evaluates multilingual capabilities across 121+ languages.
>
> Moreover, if the training data also contains annotations across these dimensions, then such signals could be directly integrated into J1 training as well by assigning fine-grained rewards for them (in addition to the coarse correctness reward in current J1 training).
>
> **4. Qualifying Statements**
> > A couple of statements (Pointwise Consistency and RewardBench SOTA) could use some qualification
>
> We thank the reviewer for these corrections.
> - First, we have explicitly written that pointwise judges are inherently “position”-consistent. As noted, they can indeed sample different scores across seeds.
>
> - Second, for Rewardbench, we have updated the text to clarify that J1 outperforms all “generative” RMs in the revised version.
>
> **5. Thinking Token Length**
> > Regarding Thinking Token Length. Why previous work [2] shows length explode to 10k, and J1’s length settles around 500 tokens? Would performance be better if the models reasoned for longer before providing a score/verdict?
>
> We believe the reviewer is referring to Figure 5 which utilizes Llama-3.1-8B-Instruct. Unlike the models used in [2] (which relies on DeepSeek-R1-Distill-Qwen, a model explicitly trained for extended reasoning), Llama-Instruct has a strong prior for concise, user-facing responses and does not naturally tend to explode reasoning length.
>
> In our experiments, we did not observe a linear correlation between extreme thinking length and judgment accuracy for this task. This echoes findings in L1 (Aggarwal et al., 2025, arXiv:2503.04697), which demonstrates that reasoning models can be optimized to produce shorter, efficient chains of thought without sacrificing performance.
>
> Consequently, we set a maximum sequence length of 4096, so even for qwen3 models our length stabilises at 900.
>
> **6. Statistical Significance**
> > In L.262, the authors state the results are statistically significant "(p<0.0001)". How was this statistical testing/resampling done?
>
> We performed a non-parametric bootstrap test by sampling (with replacement) from the large 22K evaluation set of PPE. In Appendix Table 9, we also report variance across 8 random seeds which shows that variance is negligible.

---

> > ### Comment · Reviewer_TdB2 · 2025-11-24
> >
> > I thank the authors for their detailed responses as they have addressed my concerns. I will increase my score.

---

> > > ### Author Response · Authors · 2025-12-03
> > >
> > > We thank the reviewer for the reply and recommending acceptance of our work! We are glad to hear that our response has address all the questions.

---

### Official Review · Reviewer_UxRN · 2025-11-01

**Soundness:** 3
**Presentation:** 3
**Contribution:** 4
**Rating:** 8
**Confidence:** 2

**Summary:**

The paper proposes J1, a reinforcement-learning (RL) framework to train “LLM-as-a-Judge” models to think before deciding. The key idea is to convert both verifiable (e.g., math) and typically non-verifiable judging tasks into a unified, verifiable formulation; then optimize the thinking traces and final decisions using GRPO, with rewards for verdict correctness and order-consistency to mitigate position bias. J1 is instantiated on Llama-3.x and Qwen3 (8B/32B/70B) and evaluated on PPE, RewardBench, JudgeBench, RM-Bench, and FollowBenchEval, reporting SOTA or competitive results—especially for J1-Qwen-32B-Multitask—despite training on ~22k synthetic preference pairs. The paper includes ablations (pairwise vs pointwise vs multitask; reward variants; seed prompts) and qualitative analyses of learned evaluation strategies.

**Strengths:**

- Unified, verifiable training recipe across “verifiable” and “non-verifiable” prompts enabling direct optimization via RL, not just DPO/SFT.
- Bias mitigation at training time via batching both orderings and an order-consistency reward; clean and effective.
- SOTA or competitive results on multiple judging benchmarks at reasonable model sizes, with notably strong PPE Correctness and RewardBench numbers; comprehensive comparison tables.
- Thoughtful ablations (rewards, prompts; pairwise vs pointwise vs multitask) and qualitative traces illustrating learned strategies (criteria, reference answers, self-correction).

**Weaknesses:**

- “Non-verifiable” to verifiable conversion is underspecified. For subjective tasks, the paper leans on synthetic construction and pairwise labels; the validity of these labels as ground truth for reward is not rigorously validated with humans.
- Limited causal analysis of bias reduction. While order-consistency improves, the paper doesn’t isolate the effect of batched dual-ordering vs consistency reward vs prompt phrasing with confidence intervals on all benchmarks.
- A minor detail is that while prompts and some details are provided, RL stability (seed variance, KL control, learning rate sweeps, early stopping criteria) could be reported more completely to ensure replicability of gains.

**Questions:**

- How sensitive are results to adding graded thought-quality rewards (e.g., rubric adherence, factual checks) vs the current binary signals?
- For “non-verifiable” tasks, what fraction of pairwise labels are ambiguous/tied under careful human review?
- How does J1 fare on unseen domains/languages and with adversarial perturbations (e.g., equal-quality or equal-error pairs) that should produce ties?

---

> ### Author Response · Authors · 2025-11-20
>
> Thank you for the positive review and insightful feedback. We appreciate that you found our unified recipe for achieving SOTA results promising, and our bias mitigation and ablation studies effective.
>
> Below we address your specific questions:
>
> **1. “Non-verifiable” to Verifiable Conversion and Label Validity**
>
> In Section 2.1 (lines 132-136), we discussed how non-verifiable prompts are converted to verifiable tasks. We do this by first generating a “noisy” variant of the original instruction and then producing a response to this noisy instruction. This methodology follows previous work (Wang et al., 2024d) , and we explicitly included the prompt template in Figure 14 and a concrete example in Table 5.
>
> To address the question of label validity and ambiguity, we further sampled 50 examples from this subset and labeled manually the chosen and rejected responses. We confirmed that 100% of the synthetic labels matched human judgment. As the rejected response is generated for a modified instruction, ties or ambiguity are rare.
>
> **2. Causal Analysis of Bias Reduction**
> >  Isolate the effect of batched dual-ordering vs consistency reward vs prompt phrasing
>
> We isolated these factors through the ablation studies presented in Table 8 and Table 4 of the submission:
> - **Batched Ordering vs. Consistency Reward**: In Table 8, we explicitly disentangle data augmentation from the reward signal. While training on "Both-order data" shows a moderate gain in consistent accuracy, adding the Consistency Reward has a much more substantial jump (+4.8 for PPE) and drop in verdict flips (-8.1). This causally proves the consistency reward signal itself is a major factor for the bias reduction.
> - **Prompt Phrasing**: In Table 4 we show J1 is robust to prompt variations, performing on average comparably with "Thinking" (61.8) vs. "Plan & Execution" (62.1) seed prompts
>
> Regarding confidence intervals, we show Statistical Significance in Table 9, where only negligible performance variance (±0.1) across 8 random seeds is observed. Given that the gain from Consistency Reward is orders of magnitude larger than this typical variance and also because PPE is a large test set (with 22k samples), we are confident that these results represent statistically significant causal effects.
>
> **3. Sensitivity to Reward Design (Graded vs. Binary)**
>
> We thank the reviewer for this question. While graded thought-quality rewards are an interesting direction to extend the current work, synthetically generating verifiable labels for fine-grained rubrics or factual checks is significantly more challenging than binary outcome verification. Thus the current J1 recipe primarily focuses on unifying verifiable and non-verifiable tasks into a verifiable binary reward signal to enable scalable RL. That said, the J1 training recipe can be easily extended to rubric adherence or factuality checks by deriving (binary) rewards from the satisfaction of each such criteria and then aggregating them.
>
> **4. RL Stability (seed variance, KL control, learning rate sweeps, early stopping criteria)**
>
> As detailed in Appendix C and Table 10, we controlled stability via specific hyperparameter choices:
> - KL Control: We swept KL coefficients {0.1,0.01,0.001,0} for J1-Llama-8B, selecting 0.01 as optimal. For the 70B model, we set KL to 0 to encourage exploration.
> - Entropy Bonus: We found that entropy bonuses degraded performance (leading to repetitive outputs), so we excluded them.
> - Learning Rate: We used a learning rate of 1e−6 (decaying to 3e−7 for 70B).
> - Stability: As shown in Table 9, our method is highly stable across random seed
>
> **5. Robustness: Unseen Domains, Languages, and Adversarial Ties**
>
> - **Unseen domains/languages**: First note that current J1 evaluation already demonstrates strong generalization capabilities to unseen languages and domains.
> While training on 22K English data, J1 shows strong performance on PPE Preference, which comprises 10.2K samples across 121+ languages (line 204 in the paper).
> Regarding unseen domains, J1 also generalizes well to domains like Safety (in RewardBench and PPE) and Constraint Satisfaction (FollowBenchEval) which we specifically did not train for.
>
>  - **Adversarial ties**: We conducted an experiment using 100 pairwise examples where we set both responses to be identical (Response A = Response B). Evaluating `J1-Qwen-32B-MultiTask` revealed:
>    - 99% Success Rate: The model correctly identified the tie in 99% of cases.
>    - In 92% of cases, the model explicitly stated in its thought trace that the responses were identical (e.g., "Looking at Assistant A   and B's answers: they are exactly the same... same equations, same conclusion") and assigned exactly equal scores.
>    - Of the remaining cases, 7 had a score difference <0.5 (also considered ties), and only one example had a difference of 1.
>
> This demonstrates that despite being trained strictly on preference pairs, J1 generalizes effectively to detect and handle adversarial ties.

---

> > ### Comment · Reviewer_UxRN · 2025-11-20
> >
> > I appreciate the detailed response as it addressed my concerns, so I maintain my score.

---

> > > ### Author Response · Authors · 2025-12-03
> > >
> > > We thank the reviewer for the reply and recommending acceptance of our work! We are glad to hear that our response has address all the questions.

---

### Official Review · Reviewer_dZdp · 2025-11-03

**Soundness:** 3
**Presentation:** 3
**Contribution:** 3
**Rating:** 6
**Confidence:** 4

**Summary:**

This work uses RL to train thinking LLM judges and show that it achieves SOTA performance on multiple benchmarks. Here are the contributions:
1. a recipe that converts different judgement tasks into a unified format with verifiable reward to train with RL
2. RL training on the thinking judges that shows strong performance improvement across various benchmarks
3. Develop methods to address positional bias through consistency reward.

**Strengths:**

1. Strong Empirical Results: The model trained using RL showed strong and consistent improvement across benchmarks, and is able to match frontier model (e.g. o3-mini, Deepseek-R1) that is an order of magnitude bigger.
2. Comprehensive Ablation & Analysis: The author provides thorough analysis on things like positional bias in the ablation study, which helps better understand the behavior of the model, and show that through the consistency reward, the "Verdict Flip/Ties" rate decreases.

**Weaknesses:**

1. Training Complexity: while J1 shows stronger performance than EvalPlanner, which uses offline DPO training, it is not a fully apple-to-apple comparisons. It is unclear how these two methods (GRPO vs DPO) differ under the same compute budget.

**Questions:**

1. It seems like for Qwen32B, there is a much lower improvement (e.g. +3.5 overall, +0.7 on JudgeBench) compared to the base model, whereas Llama model shows a much stronger improvement. Can the author comment on why this is the case?
2. Can the authors elaborate on the compute cost (e.g., GPU-hours, convergence time) of J1 (GRPO) versus the DPO-based EvalPlanner baseline? This would provide a more complete picture of the trade-offs involved.

---

> ### Author Response · Authors · 2025-11-20
>
> We thank the reviewer for their encouraging review, acknowledging our recipe for SOTA results and providing extensive ablation and analysis. We address the comments as follows:
>
> **1. Qwen-32B vs. Llama Improvement**
> > Qwen32B shows a much lower improvement, whereas Llama model shows a much stronger improvement
>
> We offer two explanations:
> - The models have vastly different baseline capabilities. The Llama models start from a much lower baseline (on average 48.3 for 8B and 64.3 for 70B in Table 2), offering more room for improvement. In contrast, Qwen3-32B is already a very strong baseline (77.3), making its +3.5 "Overall" gain to 80.8 significant.
> - Looking at the Table 1 breakdown of the PPE performance, we see Qwen3-32B shows much greater improvement in the challenging reasoning and coding tasks that our "thinking" optimization directly targets: for instance GPQA (Reasoning) +15.0% gain and MBPP-Plus (Coding) +10.4% gain. This also demonstrates the success of J1’s online RL recipe on complex reasoning tasks.
>
> **2. Training Complexity and Compute Cost**
> > Training Complexity/compute cost of J1 (GRPO) versus the DPO-based EvalPlanner baseline:
>
> In addition to computational resources used for our experiments indicated in our Reproducibility Statement and Appendix C (referenced inline 196), we summarize the compute cost of J1 (GRPO) below to provide further clarity,:
> - 8×A100 for J1-Llama-8B, converging in approximately 48 hours
> - 32×A100 GPUs for J1-Qwen-32B, converging in approximately 44 hours
> - 64×A100 GPUs for J1-Llama-70B, converging in approximately 80 hours
>
> We report EvalPlanner results from the original paper. Conceptually, GRPO (J1) is indeed more compute-intensive because it requires generating multiple online rollouts (N=5 in our setup). In contrast, DPO (EvalPlanner) is typically faster per training step as it optimizes on static (offline) paired data, though it shifts the compute burden to the offline data preparation beforehand.

---

### Official Review · Reviewer_Rk8V · 2025-11-13

**Soundness:** 3
**Presentation:** 3
**Contribution:** 3
**Rating:** 6
**Confidence:** 3

**Summary:**

This paper proposed J1 model which combines verifiable rewards and reinforcement learning, enabling LLM judges to develop deeper reasoning, fairer evaluation, and more consistent decision-making across diverse tasks.

**Strengths:**

The paper introduces a unified framework that converts both verifiable and non-verifiable evaluation tasks into verifiable formats using synthetic preference pairs; applies online RL to directly optimize the chain-of-thought reasoning in LLM judges; showing a novel consistency-based reward that enforces the same judgment regardless of response order; and develops a multitask J1 model that jointly learns from both pairwise and pointwise supervision. Although similar ideas have been employed in other papers, their application to LLM Judge is novel.

**Weaknesses:**

1.	The current setup focuses solely on pairwise and pointwise evaluation, without exploring extensions to multi-response or listwise judgment
2.	The work defines both Verdict Correctness and Verdict Consistency rewards, but lacks any reward weighting or sensitivity analysis
3.	The data used for training and evaluation primarily covers conversational and reasoning domains, with no experiments on diverse areas such as code or multimodal judgment
4.	This paper omits any discussion of training cost or computational resources

**Questions:**

see above weakness

---

> ### Author Response · Authors · 2025-11-20
>
> We thank the reviewer for acknowledging the novelty of J1 for LLM judges and its novel design of consistency-based rewards and multitask formulation. We address your specific comments below:
>
> **1. Scope of Evaluation: Pairwise/Pointwise vs. Listwise**
> > The current setup focuses on pairwise and pointwise evaluation, without listwise judgment
>
> Thank you for the suggestion! We chose to focus on pairwise and pointwise evaluations for three main reasons:
> - Pointwise and pairwise setups are the widely used standards for RMs and downstream evaluations respectively. This is also reflected in the benchmarks we report results on which are majority pairwise. In fact, our main benchmark, PPE, does represent listwise or “best-of-N” evals by breaking it down into multiple pairwise evals.
> - Thus we note that listwise evaluation setups can always be decomposed into pointwise or pairwise evaluation, for example by prompting the pointwise judge ‘n’ times or by constructing nC2 pairs for pairwise judges. Therefore, our current models are already capable of tackling listwise evaluation tasks.
> - J1’s reward design is such that it can directly transfer to training a listwise-J1 too, although listwise synthetic data generation is more nuanced compared to pairwise. Given that our current formulation can already be applied to listwise evaluation, we believe training a dedicated listwise-J1 is an interesting extension for future work.
>
> **2. Reward Weighting and Sensitivity Analysis**
> > reward weighting or sensitivity analysis for Verdict Correctness and Verdict Consistency rewards
>
> We address the design choices regarding weighting and sensitivity below:
> - **Reward weighting**: Note that J1’s reward modeling does not include any reweighting hyperparameter. It intentionally applies a simple unweighted sum of the verdict correctness and verdict consistency rewards, to avoid tuning additional hyperparameters.
> - **Reward Sensitivity Analysis**: We, in fact, have this result in Table 8 in appendix (discussed in lines 373-375), where we investigate the sensitivity of our model to the verdict consistency reward. Comparing the “Both-order data” and “Verdict Consistency Reward” rows, we directly ablate the weight of 0 vs. 1 for the consistency term. On both PPE and Judgebench, we observe significant improvements in Consistent Accuracy (+4.8 for PPE and +3.4 for JudgeBench) and a corresponding drop in the verdict flip ratio. This shows that J1’s training is quite responsive to the consistency signal.
>
> **3. Data Diversity (Code and Multimodal)**
> > The data used for training and evaluation primarily covers conversational and reasoning domains, with no experiments on diverse areas such as code or multimodal judgment
>
> - **Code**: Both J1 training and evaluation, in fact, does include code. As shown in Table 1, the PPE Correctness benchmark includes MBPP-Plus (Python programming benchmark). Our J1-Qwen-32B-MultiTask model shows a significant improvement (+10.4) over the base model. Furthermore, as detailed in lines 207-209, JudgeBench explicitly covers coding as one of the categories. For training, we use WildChat prompts which also includes some coding and programming queries.
> - **Multimodal**: We acknowledge that multimodal-J1 is outside the scope of this work. The base models we chose (Llama3, Qwen3) are also text-only models. However, our core J1 recipe is modality-agnostic and it would be interesting to extend the recipe beyond text in future work.
>
> **4. Training Cost and Computational Resources**
>
> As indicated in our Reproducibility Statement and Appendix C (referenced inline 196), we have detailed the computational resources used for our experiments.
>
> To provide further clarity, we summarize the compute and convergence times below:
> - 8×A100 for J1-Llama-8B, converging in approximately 48 hours
> - 32×A100 GPUs for J1-Qwen-32B, converging in approximately 44 hours
> - 64×A100 GPUs for J1-Llama-70B, converging in approximately 80 hours

---

### Meta-Review · Area_Chair_eiwK · 2026-01-05

**Summary:**

The submission received generally positive initial scores, with reviewers praising the empirical results and the clarity of the "verifiable reward" formulation. The primary concerns that shaped the recommendation were centered on the robustness, transparency, and practicality of the RL-based "Thinking Judge" framework.

**Reviewer Concerns:**

Addressed Concerns:
1. Causal Link for Consistency: The authors provided a clear ablation showing that the Consistency Reward significantly outperformed simple data augmentation.
2. Data Diversity: The authors successfully pointed to existing coding benchmarks (MBPP-Plus) and provided new experimental results on PPE Preference and adversarial ties (99% success rate).
3. Computational Transparency: The rebuttal provided granular training stats (e.g., 80 hours on 64×A100s for the 70B model), resolving the "black box" concern regarding RL costs.

Outstanding Concerns:
1. Multimodal Extension: While the authors provided a roadmap, the current J1 framework remains text-only. This is a minor limitation for a "unified" judge but does not detract from the core text-based contribution.

**Reviewer Scores:**

UxRN	8-->8	Maintained score during the rebuttal; explicitly stated concerns were addressed;
TdB2	6-->8	Increased score during the process; acknowledged that detailed responses addressed all concerns;
dZdp	6-->6/8	Likely to maintain or increase;
Rk8V	6-->6	Likely to maintain. Addressed on code/diversity, though might still feel "listwise" evaluation is a missing feature.

---

### Decision · Program_Chairs · 2026-01-26

Accept (Poster)